# An Automated Image Processing Module for Quality Evaluation of Milled Rice

**DOI:** 10.3390/foods12061273

**Published:** 2023-03-16

**Authors:** Chinmay Kurade, Maninder Meenu, Sahil Kalra, Ankur Miglani, Bala Chakravarthy Neelapu, Yong Yu, Hosahalli S. Ramaswamy

**Affiliations:** 1Department of Mechanical Engineering, Indian Institute of Technology, Jammu 181221, India; 2College of Biosystems Engineering and Food Science, Zhejiang University, 866 Yuhangtang Road, Hangzhou 310058, China; 3Department of Mechanical Engineering, Indian Institute of Technology, Indore 453552, India; 4Department of Biotechnology and Engineering, National Institute of Technology, Rourkela 769008, India; 5Department of Food Science, McGill University, 21111 Lakeshore Road, St-Anne-de-Bellevue, QC H9X 3V9, Canada

**Keywords:** automation, computer vision, Raspberry-Pi, quality assessment, machine learning, rice grains

## Abstract

The paper demonstrates a low-cost rice quality assessment system based on image processing and machine learning (ML) algorithms. A Raspberry-Pi based image acquisition module was developed to extract the structural and geometric features from 3081 images of eight different varieties of rice grains. Based on features such as perimeter, area, solidity, roundness, compactness, and shape factor, an automatic identification system is developed to segment the grains based on their types and classify them by using seven machine learning algorithms. These ML models are trained using the images and are compared using different ML models. ROC curves are plotted for each model for quantitative analysis to assess the model’s performance. It is concluded that the random forest classifier presents an accuracy of 77 percent and is the best-performing model for the classification of rice varieties. Furthermore, the same algorithm is efficiently employed to determine the price of adulterated rice samples based upon the market price of individual rice.

## 1. Introduction

Rice is the most extensively consumed cereal around the globe [1]. The quality of the rice grain has a considerable impact on both the yields of rice for farmers and its economic return. Several improved rice varieties have been developed in past few decades to meet the demand of consumers for high-quality rice [2]. However, seasonal and geographical variations result in significant variation in yield, physical and nutritional quality of rice of same variety which in turn affects the market value of rice [3,4,5]. Therefore, rice is extremely vulnerable to fraud around the world [1]. There are four different types of rice adulteration observed in the market: (a) substitution with look-alike materials of low cost, (b) substitution with low-quality rice grains, (c) dilution of the original product, and (d) mislabeling of the age and origin of the material [6,7]. This adulteration can incur significant economic loss for food companies and consumers. The commercial value of the grains such as rice primarily depends on their chemical composition and structural features. Their chemical constituents are determined using various lab procedures that take time, raise costs, have a certain ecological footprint and need specific knowledge. Furthermore, vibrational spectroscopy in conjunction with chemometrics has also been employed to determine the chemical composition and varietal classification of various food grains [8,9,10,11]. These methods are non-destructive and eco-friendly, but require expensive instruments. The structural features of rice grains involve the determination of geometric features (size, shape) and other traits such as color, chalkiness, morphological and textural features. These physical features are generally visible to the naked eye and can be measured manually. However, the process of manual inspection is quite laborious, inconsistent, subjective, and time-consuming [12]. Near infrared spectroscopy and Fourier-transform infrared spectroscopy are also efficiently employed for adulteration detection in various food materials with minimal sample preparation [13,14,15,16]. In addition, near-infrared hyperspectral imaging has also been used for accurate classification and quantification of adulteration of different foods [17,18,19,20]. These features can also be detected via computer vision and can be automated easily. These computer vision techniques are efficient, non-destructive, less time-consuming and can also be efficiently employed at an industrial scale for on-site detection of rice adulteration [21,22]. Furthermore, with advancements in computer science, mechanical and automation engineering, there is huge scope for image processing and computer vision for food quality assessment based on machine learning [23]. Several studies have successfully employed digital image processing (DIP) techniques for feature extraction, classification and quality prediction of various foods materials [24]. DIP-based food quality assessment is a fast, non-invasive, non-destructive, safe, energy-efficient, low-cost technique that does not require skilled personnel to operate the instrument. Previously, researchers have successfully employed DIP for fractal analysis of retrogradation of rice starch [25] and authentication of rice varieties as well as the derived flour [6,26]. In these studies, the individual rice grains were segmented from the background and their extracted geometric parameters were used to classify the rice grains as either high, medium or low quality. In the present study, the grains are classified using image processing algorithms and machine learning algorithms. Rice grain segmentation and feature extraction are performed automatically by image processing algorithms, and these features are fed to machine learning algorithms to automatically classify, which makes the system fully automatic. In addition, previously, researchers have employed expensive cameras and smartphone cameras as image capturing devices [24], which ultimately adds to the cost of overall setup. However, in the present study, the Raspberry-Pi module used is a low low-cost, portable and easy-to-use device that resulted in an overall setup cost of only USD 50. Thus, the primary objective of the study is to develop an efficient, fully automated and cost-effective system to classify rice grains based on varieties such as Basmati, Eco Kolam, HMT Kolam, Kana Basmati, Sona Masuri, Tibar Basmati, Tukda Basmati, and Wada Kolam. This study explored different ML algorithms such as logistic regression (LR), decision tree (dT), random forest (rF), multilayer perceptron (MLP), and support vector machine (sVm) with linear, polynomial, radial basis function (RBF), and sigmoid kernels.

## 2. Materials and Methods

### 2.1. Raspberry-Pi-Based Machine Vision System

The low-cost Raspberry-Pi module is used for the classification of different types of rice grains (Figure 1a). This module is equipped with a 16 GB memory card that acts as a primary disk for operating the Raspberry-Pi desktop system–Debian version 10 (buster) and the storage medium for captured rice images. Figure 1b shows the steps that are followed to achieve the rice classification using the Raspberry-Pi module-based machine vision system: image acquisition, pre-processing, segmentation, feature extraction, and the training and testing of machine learning models for classification.

### 2.2. Sample Collection

Eight different varieties of rice grains commonly consumed in the region, namely Basmati (BM), Kana Basmati (KB), Tibar Basmati (TB), Tukda Basmati (TKB), Eco Kolam (EK), HMT Kolam (HK), Wada Kolam (WK), and Sona Masuri (SM), were collected from the local market, Mumbai, India, in 2020. Appendix A details the local market price and the number of samples for each rice variety. The majority of collected rice grains were healthy, except for a few, in which partial chalkiness was observed in the grains.

### 2.3. Imaging System

The rice grain images were acquired using a 5 MP integrator IR-Cut camera (OV5647 5MP 1080P) by Omnivision Technologies (Santa Clara, CA, USA) with a 0.25-inch CCD sensor and an adjustable focal length. The IR-cut filter helps in reducing the color distortions resulting from IR light during daylight. The camera was interfaced with Raspberry-Pi (Raspberry-Pi foundation group, Cambridge, UK) using a flat 15-pin Camera Serial Interface (CSI) ribbon cable for power and the relay. The operating system raspi-config is installed in the Raspberry-Pi module. For starting the camera and image acquisition, Python script was used from the host computer (Dell.Inc, i7 processor, 64 GB Ram and Nvidia GPU) using the Secure File Transfer Protocol (SFTP) and stored in JPEG format. All rice grain images were taken against a dark blue background at a fixed distance from the camera (Figure 1a). The blue background color is preferred over a black background as it provides a better contrast, which enables easy identification of grey and black colored objects such as soil or rock particles. The complete experimental setup was placed in a closed chamber, and multiple LED lights were placed at uniform distance to increase illumination.

### 2.4. Dataset Details

The image dataset consists of 3081 images with 230 images (approx.) acquired for each variety (Appendix A). For each image, randomly selected 80 grains are spread uniformly without touching each other. Each image is in RGB format with a resolution of 2592 × 1944 pixels. For analyzing each rice grain, image segmentation is performed (Section 2.5.1). A 5 × 5 median filter is used to remove salt and pepper noise resulting from stray reflections from surroundings. This filter considers a 5 × 5-pixel region around a particular pixel and replaces it with the median value for RGB channels.

### 2.5. Image Post-Processing

#### 2.5.1. Image Segmentation

Image segmentation is performed to isolate each rice grain to enable its feature extraction. The original RGB image is converted to a grayscale image and the individual rice grains are identified against the background using Otsu’s thresholding technique [27]. Subsequently, the Watershed algorithm is used to extract the foreground and the background, using markers to detect the boundaries. Otsu’s thresholding algorithm was obtained from Open-CV library. The procedure for determining the threshold value involves computation of a histogram from a grayscale image and the probabilities of occurrence of all the $k^{th}$ intensity levels using Equation (1):(1)$$p\left(k\right)=\frac{\mathrm{Number}\;\mathrm{of}\;\mathrm{pixels}\;\mathrm{with}\;\mathrm{intensity}\;k}{Total\;number\;of\;pixels}$$

The histogram for a representative image is shown in Appendix A.

Second, the initial class probability is calculated as
(2)$$U_{o}\left(t\right)=\sum_{k=t_{min}}^{t-1}p\left(k\right);\;\mathrm{and}\;U_{1}\left(t\right)=\sum_{k=t}^{t_{max}}p\left(k\right)$$
where, $U_{i}$ are the class probabilities with $U_{o}$(0) = 0, and $U_{1}$(0) = 1

Third, the class mean is calculated as
(3)$$m_{o}\left(t\right)=\sum_{k=t_{min}}^{t-1}\frac{kp\left(k\right)}{U_{o}\left(t\right)};\;\mathrm{and}\;m_{1}\left(t\right)=\sum_{k=t_{min}}^{t-1}\frac{kp\left(k\right)}{U_{1}\left(t\right)};$$
where $m_{i}$ is the class means (classes being 0 and 1 for binary), initially, $m_{o}\left(0\right)=$ 0 and $m_{1}\left(0\right)=$ 1. The parameters t  and k represent the pixel intensity and threshold intensity levels, respectively.

Subsequently, the interclass variance is calculated as
(4)$$V_{b}^{2}\left(t\right)=U_{o}\left(t\right)U_{1}\left(t\right){\left[m_{o}\left(t\right)-m_{1}\left(t\right)\right]}^{2}$$

If $V_{b}^{2}\left(t\right)>V_{b}^{2}max$
$$V_{b}^{2}\left(t\right)max=V_{b}^{2}\left(t\right),\mathrm{and}\;\mathrm{thresh}=t$$
where $V_{b}^{2}\left(t\right)$ is the interclass variance and $V_{b}^{2}\left(t\right)max$ is the maximum value for which the search is performed to obtain the optimum threshold value *thresh*. The pixel intensity varies from 0–255.

The final output of threshold is the intensity value corresponding to maximum variance $V_{b}^{2}\left(t\right)max$ by which the binary image is generated. The original image and the result after thresholding is shown in Appendix A, respectively. A representative example of image thresholding with 150 rice grains is shown in Appendix A for different values of the threshold intensity, along with the corresponding values of class-mean, variance, and threshold.

Next, the segmentation of each rice grain is carried out using marker-based Watershed algorithm [27]. In this algorithm, the marker is generated via morphological erosion operation on the binary image. Erosion removes the noisy pixels, smooths the object boundaries, and removes the outer layer of object pixels. It consists of an input image and a structuring element, which removes the boundary pixels from image depending on the degree of overlap. A 3 × 3 structuring element is used to perform the operation to ensure that the rice grains present in proximity are separated as two different grains and assigned with different markers. Subsequently, a connected component analysis is performed to obtain labels for all the markers. The resulting binary images are then passed through the Watershed algorithm to generate labels for the markers, which completes the segmentation process. The labelling of grains is represented by a unique random colored mask, and the images after thresholding and segmentation are shown in Appendix A, respectively. After segmentation, the images are stored as individual rice grain images in JPEG format. Note that the Watershed algorithm fails to separate the grains that touch each other (Appendix A), and therefore, these are removed from the image dataset.

#### 2.5.2. Feature Extraction

In this step, four different types of features, namely, the geometrical and morphological features, color features, and textural features are extracted from the individual rice grain images to train the machine learning models. The methodology for extraction of each type of feature is detailed in the following sections.

#### 2.5.3. Geometrical and Morphological Features

The perimeter, area, solidity, roundness, compactness, and shape factor are extracted from the individual rice grain images using Open CV library [20]. A complete list of all the extracted features along with their mathematical expressions is detailed in Appendix A. First, the contour (Appendix A) is generated for each rice grain by applying edge detection on binary image. This contour represents the boundary that encloses the regions with same pixel intensity. The region bounded by the contour represents the 2D-projected area of grain while the contour length represents the grain perimeter. These geometrical parameters are used to determine equivalent diameter, and other morphological features such as the roundness, and compactness of each grain. Subsequently, an ellipse that best fits the contour is constructed around the rice grain to determine is length (major axis), width (minor axis), and aspect ratio. The length and width of rice grains are approximated by the length of major and minor axis of ellipse, respectively. The length, width, and aspect ratio of a few representative rice images are shown in Appendix A. Additionally, a bounding box/rectangle is fitted to the contour to determine morphological features, namely the spatial extent and the shape factor of rice grains.

#### 2.5.4. Color-Based Features

The color-based features, namely, the Hue, Saturation and Value (HSV) are extracted for region inside the bounding box of a rice grain. HSV is an alternate representation of RGB model that is closely aligned with the way humans perceive color-making attributes: *H* describes the pure color, *S* measures the degree to which the pure color is diluted, and *V* represents the intensity that describes relative brightness of color. The R, G, B values are converted to HSV values using Equation (5):

Intensity,
(5)V=M=max(R,G,B)
m=min(R,G,B)

Saturation, $S=\frac{M-m}{M}$, if M>0 or S=0  if M=0

Hue, $H=60\times \left(0+\frac{G+B}{M-m}\right),\;$ if M=R
$$H=60\times \left(2+\frac{B-R}{M-m}\right),\;\mathrm{if}\;M=G$$
$$H=60\times \left(4+\frac{R+G}{M-m}\right),\;\mathrm{if}\;M=B$$
H=H+360 if H<0

Another color-based feature is the histogram, which is a graphical representation of the distribution of intensity levels in each of the six channels (*R*, *G*, *B*, *H*, *S*, *V*) of the image. A histogram plots the number of pixels corresponding to each intensity level. For each segmented rice grain image, six histograms are plotted, i.e., one for each channel (Appendix A). Since the intensity value ranges from 0–255, a total of 256 values are possible for a single channel. However, the intensity levels of original image are quantized from 256 to 48 to obtain a histogram of 48 bins from each of the six channels to reduce computational time. A total of 288 (48 × 6) histogram features are calculated as follows:(6)$$hist\;\left(i\right)=\sum_{x=1}^{N}\sum_{y=1}^{M}\{\begin{array}{c} 1,\;if\;I\left(x,y\right)=i\\ 0,\;otherwise \end{array}\;i\in \left[1,48\right]$$

#### 2.5.5. Textural Features

The texture of rice grains determines the hardness, adhesiveness, cohesiveness, gumminess, springiness, resilience, and chewiness of cooked rice. For a rice grain image, the texture is measured in terms of spatial arrangement of intensity relative to the neighborhood of any given pixel, and other factors such as fineness/coarseness, smoothness, granulation, randomness, and irregularity. In this study, the gray level co-occurrence matrix (GLCM) is used to determine textural features of the image. It provides the distribution of co-occurring intensity values at a given offset by calculating how often a given pixel with intensity value i occurs adjacent to a pixel with intensity j. Let I (x,y) be the intensity of a pixel at location (x,y) in the image; the GLCM matrix G is defined as
(7)$$G\;\left(i,j\right)=\sum_{x=1}^{N}\sum_{y=1}^{M}\{\begin{array}{c} 1,\;\;if\;I\;\left(x,y\right)=i\;and\;I\;\left(x+d_{1},\;y+d_{2}\right)=j\\ 0,\;otherwise \end{array}$$
where $d_{1}$ and $d_{2}$ are the pixel offset distances, as shown in Appendix A, and (N,M) is the image resolution (width, height). Four angles are selected for analysis: 0°, 45°, 90°, and 135°, and the values of $d_{1}$ and $d_{2}\;$ corresponding to these angles are listed alongside these directions in the coordinate frame (Appendix A). The shape of GLCM depends on the maximum intensity level and the resolution. In this study, the shape of GLCM is 256 × 256, because there are 256 intensity levels. The textural features are defined using contrast (con), and statistical features such as correlation (corr), homogeneity (ho), angular second moment (asm), energy (en) and dissimilarity (dis) at different orientations of rice. These parameters are considered as the dataset in machine learning to segregate rice based upon its quality.

#### 2.5.6. Statistical Features Using GLCM

After GLCM is derived, five statistical features, namely contrast, dissimilarity, correlation, homogeneity, energy, and angular second moment are calculated using the scikit-image Python library [20]. These statistical features are described as follows:

***Contrast (CON)*** defines the intensity difference between a pixel and its surroundings over the entire image. Therefore, for an image with a constant intensity value, the contrast has zero value. The contrast is calculated as
(8)$$CON=\sum_{i,j}^{N_{G}}{\left(i-j\right)}^{2}p\left(i,j\right)$$
where, $N_{G}\;$ is the number of intensity levels in the image.

***Correlation (CORR)*** is a measure of how closely a pixel is correlated to its neighboring pixels over the entire image. Its magnitude ranges from −1 to 1, and it is described by the following expression:(9)$$CORR=\sum_{i,j}^{N_{G}}\frac{\left(1-\mu i\right)\left(1-\mu j\right)p\left(i,j\right)}{\sigma_{i}\sigma_{j}}$$
where μi and μj  are the mean values, and $\sigma_{i}$ and $\sigma_{j}$ are the standard deviations. These are expressed as follows: $\mu_{x}=\sum_{i=1}^{N_{G}}\frac{\rho_{i}^{x}}{N_{G}}$ and $\mu_{y}=\sum_{j=1}^{N_{G}}\frac{\rho_{i}^{y}}{N_{G}}$; $\sigma_{x}^{2}=\sum_{i=1}^{N_{G}}\frac{{\left(\rho_{i}^{x}-\mu_{x}\right)}^{2}}{N_{G}}$ and $\sigma_{y}^{2}=\sum_{j=1}^{N_{G}}\frac{{\left(\rho_{i}^{y}-\mu_{y}\right)}^{2}}{N_{G}}$.

Here, $\rho_{i}^{x}$ and $\rho_{j}^{x}$ are the marginal probability distributions, which are given by $\rho_{i}^{x}=\sum_{i=1}^{N_{G}}G\;\left(i,j\right)$ and $\rho_{j}^{x}=\sum_{j=1}^{N_{G}}G\;\left(i,j\right)$.

***Homogeneity (HO)*** represents the closeness of the distribution of elements in GLCM along the GLCM diagonal, and is given by
(10)$$HO=\sum_{i,j}^{N_{G}}\frac{p\left(i,j\right)}{1+\left|i-j\right|}$$

***Angular Second Moment (ASM)*** is a sum of the squares of all elements of GLCM, and is calculated using the following equation:(11)$$ASM=\sum_{i=1}^{N_{G}}\sum_{j=1}^{N_{G}}G{\left(i,j\right)}^{2}$$

***Energy (EN)*** is the square root of the angular second moment, and is given by
(12)$$EN=\sum_{i,j}^{N_{G}}p{\left(i,j\right)}^{2}$$

***Dissimilarity (DIS)*** is also a measure of the difference in the intensity levels between the pixels. However, unlike contrast, dissimilarity is linearly weighted. It is given by the following expression:(13)$$DIS=\sum_{i=1}^{N_{G}}\sum_{j=1}^{N_{G}}\left|i-j\right|G\;\left(i,j\right)$$

A segmented rice grain image and its corresponding maps of the six statistical features are shown in Appendix A.

#### 2.5.7. Feature Dataset

For each segmented rice grain image, a feature dataset consisting of 324 features is constructed, including 18 geometrical and morphological features, 282 color features, and 24 textural features (six texture features in four angular directions). The complete feature set is summarized in Appendix A. The features are stored in a tabular format with M rows and N columns, where each row corresponds to a unique rice grain image, and features corresponding to each image are stored as 1×N dimensional vector, i.e., N features per rice grain image. This way, the shape of the feature dataset is M×N = 3081 × 324.

### 2.6. Experiments

#### 2.6.1. Model Training

The training of model is performed using the collected image dataset, and the final model is tested using remaining data samples to predict the type and amount of adulteration present in a rice sample. To divide the data, a stratified *k*-fold cross-validation approach is followed, where the dataset is split into *k* equal parts; one part is used for testing the model, and the rest, *k*−1, are used for training. This process is repeated *k* times, and an average of evaluation metrics (i.e., the accuracy and F1 score) is calculated for *k* iterations. The *scikit-learn* Python library is used for implementing ML algorithms.

#### 2.6.2. Machine Learning (ML) Models

To enable classification of rice grains based on their variety, eight different ML models are adopted, namely, logistic regression (LR), decision tree (DT), random forest (RF), multilayer perceptron (MLP), and support vector machine (SVM) with linear, polynomial, radial basis function (RBF), and sigmoid kernels. For brevity, the RF and DT models are discussed, as the former is the best-performing model; however, for the remaining models, only results are presented.

#### 2.6.3. Decision Tree (DT) Classifier

The DT model is used for predictive analysis. At each level (node), DT classifies (splits) the data based on a certain threshold of a particular feature that is used for training. The threshold and feature for the threshold are selected to minimize the optimizing parameter (entropy or Gini-impurity). Since this is a relatively simpler classification problem, a lightweight model is chosen, which minimizes the prediction time and maintains accuracy. The DT classifier has lowest inference time compared to other tree-based classification algorithms, as it only depends on depth of DT. Based on the classification task, certain features are more relevant than others, and developing the model based on these features increases the overall accuracy of the model. This is achieved by preventing overfitting of the model on training data. On the other hand, the decreased number of features means less training time is required. The *Scikit-learn* implementation of the DT model also provides the feature importance of all the sample features. The feature importance value ranges between 0 and 1, specifying the relative importance of each feature. The number of features is selected by training the model for various numbers of features and selecting the smallest number of features that gives the desired accuracy. Here, the top 59 features are selected, based on a threshold of the feature importance value. This threshold is decided using an iterative process to optimize the model accuracy. The top 40 features are shown in the plot in Appendix A. It can be observed that majority of the features correspond to RGB or HSV channels.

#### 2.6.4. Random Forest (RF) Classifier

This is a type of ensemble learning technique whose most basic unit is a DT, which is used for predictive analysis. At each level (node), the DT classifies (splits) data based on a certain threshold of a particular feature used for training. The threshold and feature for the threshold are selected to minimize the optimizing parameter (entropy or Gini-impurity). However, single decision trees tend to overfit the training data and have a lower testing accuracy. Hence, the RF classifier which uses bagging (bootstrap aggregation) is used to provide regularization and introduce some randomness into the model. The RF classifier consists of a certain number (usually 100+) of DT classifiers. Each DT makes its own prediction, and the final prediction in the case of the classification task is the mode of all the predictions. The python library ‘scikit-learn’ is used to train the RF classifier [20].

The RF classifier has several hyperparameters, such as (a) number of decision trees, (b) depth of each DT, (c) the minimum number of samples required to split an internal node, and (d) the minimum number of samples, which are to be tuned to improve the accuracy of a model. The grid search method is used to perform hyperparameter tuning, in which a combination of all the values of hyperparameters in each range are used to determine the best combination.

## 3. Model Comparisons and Discussion

### 3.1. Performance Metrics

The key objective of this study is to explore a ML model that enables accurate, rapid and cost-effective classification of rice grains based on their variety. This is a multi-class classification problem in which comparison of performance, accuracy, precision and macro-averaged F1-scores of the models are compared. The accuracy score is analyzed by using a confusion matrix, which predicts the performance of a model for a given rice variety. For each sample in the test dataset, the confusion matrix indicates the actual class of the rice sample as well as the class of the sample predicted by the respective ML model. Precision indicates the correctness of prediction of the class, and recall indicates the efficiency of a model to predict the samples with their actual class. Precision and recall are mathematically expressed as:(14)$$Precision=\frac{True\;Positive}{True\;Positive+False\;Positive}$$
(15)$$Recall=\frac{True\;Positives}{True\;Positives+False\;Negatives}$$

The precision–recall trade-off depends upon the predicted probability threshold parameter. Hence, F1-score is used as a metric (see Equation (16)) which determines the overall performance and is obtained from the harmonic mean of the precision and recall. The performance metrics of each of the models are detailed in Table 1 for the image dataset analyzed in this study.
(16)$$F1\;score=\frac{2\times Precision\times Recall}{Precision+Recall}$$

The results in Table 1 shows that SVM with RBF kernel has the best accuracy score of 0.773. However, the accuracy score is influenced by a class having higher support, i.e., a higher number of samples in the test set. A model can have higher accuracy by correctly classifying classes with higher support even if it fails to classify classes with lower support. Additionally, since the target of the model is to determine the adulteration in the rice sample, which may be present in imbalanced amount (the precision and recall), hence the F1 score takes precedence over the accuracy score. The F1 score treats all the classes equally, especially in case of imbalanced dataset, and provides a more reliable performance indicator. Hence, the RF classifier is chosen as the best classifier for segregation of rice varieties in an adulterated sample, achieving the best F1-score of 0.761.

### 3.2. Receiver Operator Characteristic (ROC) Curves

ROC curves are plotted for all eight varieties of rice samples and analyzed with different machine learning algorithms MLP, RF, LR, SVMRBF, DT, SVM linear, SVM polynomial and SVM sigmoid. Figure 2 shows the ROC curves of eight machine learning algorithms. Since ROC curves can only be obtained for binary classification problems, we have used one vs. the rest classifiers to obtain the ROC curves for each rice type. The ROC curves helped us choose the best algorithm for the classification task, and to indicate the performance of a particular model with respect to a given class. ROC curve plots the true-positive rate (TPR) of particular class of rice type against the false-positive rate (FPR) for a particular threshold value. Here, the threshold value ranges from 0 to 1 and signifies the minimum value of probability output for a sample to belong to a respective class. For a classifier, the TPR has to be closer to 1, and the FPR has to be closer to 0. However, there must be a trade-off between both. Hence, the ROC curve closest to the point (0,1) is preferred. For the rice types of HK, KB and TB, all the models, especially RF, logistic regression and SVMRBF, perform considerably well. However, logistic regression is better for classifying BM, and RF is better for classifying TKB.

### 3.3. Rice Variety Classification Using the RF Classifier

The results for the best classification model, the RF classifier, have been demonstrated. Stratified k-fold cross-validation is carried out with *k* = 10 splits in the dataset. The highest accuracy obtained is 0.809, and the lowest accuracy obtained is 0.745. The RF model shows an average accuracy of 0.770 in classifying eight different types of rice into their respective classes. The confusion matrix and the performance metrics are given in Appendix A and Table 2, respectively. The highest precision of 0.949 is obtained for BM. From the confusion matrix, we see that three of the predicted ‘Basmati’ rice types are actually ‘Tibar Basmati’. The lowest precision of 0.564 is obtained for the class ‘HMT Kolam’. The confusion matrix indicates that what the model predicted as ‘HMT Kolam’ has a good amount of ‘Wada Kolam’ mixed in. This can be attributed to their similarity in features, due to their same overall type, ‘Kolam’. ‘HMT Kolam’ has the lowest recall value of 0.590, which, according to the confusion matrix, indicates that it has features similar to ‘Tukda Basmati’ and ‘Wada Kolam’. The highest recall value of 0.908 is obtained for ‘Kana Basmati’, which indicates that the model can distinguish it from the rest of the rice types better than any other rice type. The F1 score, which is the harmonic mean of the precision and recall values, gives us a better understanding of the overall efficiency of the model in distinguishing different rice types. The highest F1 score of 0.923 is obtained for ‘Kana Basmati’, and the lowest of 0.576 is obtained for ‘HMT Kolam’. The average F1 score obtained is 0.761.

### 3.4. Validation of the RF Classifier

The purpose of validation is to determine the efficiency of the model to detect rice adulteration. The validation is performed using four different varieties of rice samples. For the validation, 50 grains of high-quality ’Basmati’ rice variety are mixed with 30 grains of low-quality Tibar Basmati (TB) variety. Subsequently, an image is captured and results are obtained. Each of the four samples consist of higher-priced and lower-priced rice grains in a fixed ratio of 5:3. This is keep consistent to ensure fair comparison of the model’s performance on each of the four samples. The validation results of the four rice samples are detailed in Appendix A. Additionally, the average price factor per grain is calculated based on the price of each predicted rice grain. Since we do not know the weight of each grain, we will use the average length of each type to obtain the price factors. This is compared with the actual price factor of the mixture, and the price factor without adulteration for each sample. This also indicates whether the model is able to determine the actual price of the grains:(17)$$\mathrm{Actual}\;\mathrm{price}\;\mathrm{factor}=\sum p_{i}L_{i}l_{i}^{2}N_{i}$$
(18)$$\mathrm{Predicted}\;\mathrm{price}\;\mathrm{factor}=\sum p_{i}L_{i}l_{i}^{2}M_{i}$$
(19)$$\mathrm{Price}\;\mathrm{without}\;\mathrm{adulteration}\;\mathrm{factor}=\mathrm{Price}\;\mathrm{of}\;\mathrm{grain}\;\mathrm{with}\;\mathrm{highest}\;N_{i}\times L_{i}l_{i}^{2}\times \sum N_{i}$$
where $N_{i}\;$ is the number of rice samples of $i^{th}$ type, $L_{i}$ is the length of the rice sample of $i^{th}$ type, $l_{i}$ is the width of the sample, $M_{i}\;$ is the number of predicted rice samples, and $p_{i}\;$ is the price per kg of rice sample. The quality of the rice detection rate indicates that the model can identify the adulteration in the rice samples. From the validation results, it may be concluded that even though the model is not able to accurately estimate the grains, it can estimate the percentage of rice of higher or lower quality with good accuracy. Additionally, there is a significant difference between the predicted price factor and price factor without adulteration. This helps to indicate the presence of adulteration in the form of mixing of lower-quality grains with high-quality grains. The price is also major factor in determining the quality of the rice samples. The model can predict the actual price of rice with a maximum of 15% error for the four samples.

### 3.5. Qualitative Comparison of ML Models

Appendix A shows the qualitative analysis of the eight machine learning algorithms used for the assessment of rice quality estimation for Sample 1. This helps to visualize the classification performance of all the machine learning models with respect to classifying adulteration in form of lower-quality rice. Here, in Sample 1, 50 grains of ‘Basmati’ (the higher-quality rice), and 30 grains of ‘Tibar Basmati’ (the lower-quality rice) are considered. It may be observed that most of the models are able to identify the lower-quality rice; however, other rice varieties are also predicted by these classifiers due to the structural similarity of the two varieties. It may be deduced from these figures that for the RF classifier, the misclassification rate is lowest, and the F1 score of this algorithm is better than that of other classifiers.

## 4. Conclusions and Future Scope

In this study, an image processing module using Raspberry-Pi was successfully designed to detect adulteration of either foreign particles or mixing of low-quality rice with high-quality rice. The images of mostly non-touching grains were captured, and the individual grains were segmented using the Watershed algorithm. Several features were extracted based on geometry, morphology, color, and texture. Subsequently, eight machine learning-based models were used to classify the rice type, and their performance is compared in terms of accuracy, precision, recall and F1-score. The RF classifier is found to be best-performing algorithm, with a model accuracy (F1-score: 0.761) of 76.1%. To improve the current accuracy, training on different deep learning models such as Efficient Net, Inception V3, Res Net, and Mobile Net may be carried out. Reinforcement learning or online training-based approaches may also be tried to improve the model while it is in use. In future, the same rice varieties grown in different regions or years need to be included in the sample pool, as do samples of same varieties grown in a different year or region and exposed to different seasonal or environmental conditions. Inclusion of these samples during training will enhance the efficiency of models. Furthermore, a user interface, such as an Android app or a web app, may be developed to enable remote access and easy viewing of the results. Moreover, big data can be used here to train the machine learning models and assist image enhancement, image classification and segmentation, leading to an increase in model efficiency. The model may further be launched with a mobile application to evaluate the actual price of adulterated rice available in local markets.

## Figures and Tables

**Figure 1 foods-12-01273-f001:**
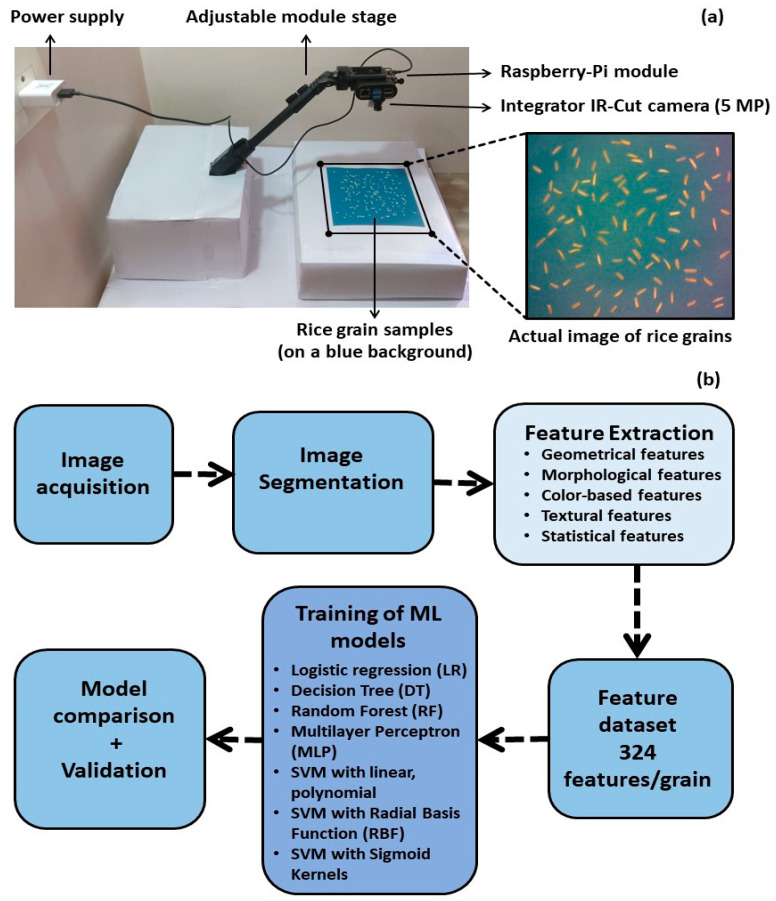
(**a**) Raspberry-Pi-based image acquisition module with integrated IR-cut camera. Actual RGB image of the rice grains is shown on the right. (**b**) Flowchart showing the algorithm used for the classification of rice grains and ascertaining their quality.

**Figure 2 foods-12-01273-f002:**
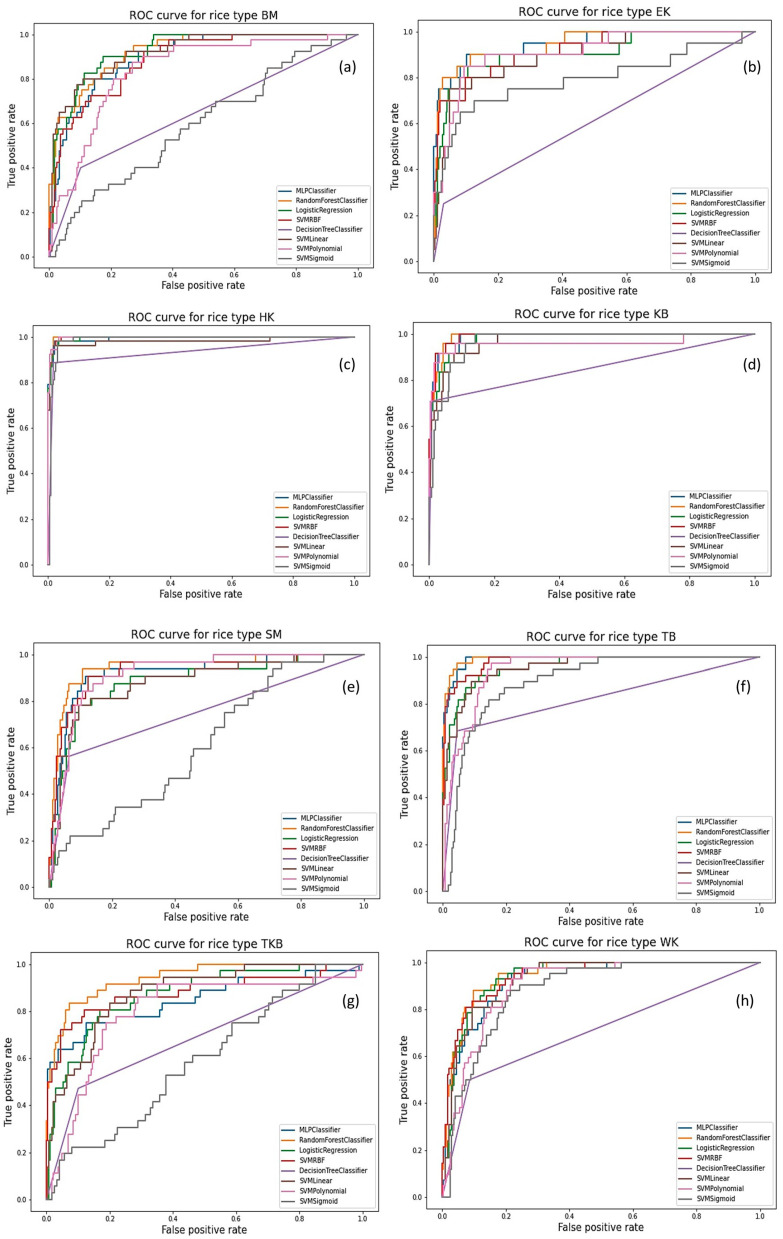
Receiver operating characteristic ROC curve for (**a**) Basmati (BM), (**b**) Eco Kolam (EK), (**c**) HMT Kolam (HK), (**d**) Kana Basmati (KB), (**e**) Sona Masuri (SM), (**f**) Tibar Basmati (TB), (**g**) Tukda Basmati (TKB), (**h**) Wada Kolam (WK) varieties of rice.

**Table 1 foods-12-01273-t001:** Performance metrics of eight machine learning models for classification of rice grains.

Performance Metric	MLP	RF	LR	DT	SVM RBF	SVM Linear	SVM Polynomial	SVM Sigmoid
Accuracy	0.734	0.771	0.768	0.676	0.773	0.764	0.705	0.699
Precision	0.730	0.767	0.739	0.646	0.762	0.742	0.728	0.718
Recall	0.728	0.760	0.739	0.655	0.751	0.748	0.683	0.669
F1-score	0.728	0.761	0.738	0.649	0.753	0.743	0.693	0.678

MLP, multilayer perceptron; RF, random forest; LR, logistic regression; DT, decision tree; SVM, support vector machine; RBF, radial basis function.

**Table 2 foods-12-01273-t002:** Class-wise performance metrics of random forest classifier.

Performance Metric	BM	EK	HK	KB	SM	TB	TKB	WK
Recall	0.888	0.785	0.590	0.908	0.786	0.796	0.674	0.746
Precision	0.949	0.634	0.564	0.939	0.843	0.746	0.666	0.784
F1-score	0.917	0.701	0.576	0.923	0.813	0.770	0.670	0.765

BM, Basmati; EK, Eco Kolam; HK, HMT Kolam; KB, Kana Basmati; SM, Sona Masuri; TB, Tibar Basmati; TKB, Tukda Basmati; WK, Wada Kolam.

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
