# Peer review of "An Automated Image Processing Module for Quality Evaluation of Milled Rice"

_foods, 2023, doi:10.3390/foods12061273_

Round 1

Reviewer 1 Report

Authors done good work. Dataset details must be given while details of models should be furnished. Also, Raspberry pi GPIO pins needs to be define and camera should be furnished in depth such as FPS.

Author Response

Response to Reviewer 1:

Thank you for your time and valuable suggestion. Your suggestions have been incorporated in manuscript and highlighted with yellow color.

Comment 1: Line 15, replace is with was.

Response: Revised in abstract as red text.

Comment 2: Line 20, which are different performance metrics

Response: Revised in abstract as red text.

Comment 3: Line 23. How price of adulterated rice samples is determined?

Response: Revised in abstract as red text.

Comment 4: Line 53, delete cost-effective

Response: Deleted.

Comment 5: Line 98-104, which one was POWER GPIO and which one was trigger pin ? how relay was activated using 3 V, if used?

Response: Dear reviewer, here Raspberry pi is used as a POWER GPIO and Python codes are written to trigger the camera module.

Comment 6: Table 2

Response: Table 2 is added to supplementary Table 5, and is presenting confusion matrix for the Random Forest classifier

Comment 7: Line 407 Basmati rice variety

Response: Correction has been done.

Comment 8: Line 452, F1-score of 0.761: dataset description missing...

Response: Correction has been done.

Reviewer 2 Report

Revised manuscript describes an artificial vision system to assess quality of rice grain. The system is formed by a digital camera controlled from a Raspberry Pi miniature computer and a machine learning identification algorithm, capable of identifying eight rice varieties, in a clear use of intelligent system against food fraud. The final algorithm for identification was Random forests, chosen by comparison from a set of seven different algorithms. Final performance achieved is an identification accuracy of 77.0%; a similar algorithm can be used for determining the value price of adulterated rice samples. Application described is interesting :

1.    Abbreviations:  a good idea may be to list the abbreviations in alphabetical order

2.    Introduction needs to comment on Near-infrared methods, which are essentially equivalent to a camera based technique, and can work by direct observation of the solid sample.

3.    Figures incorporating text (Figure 1, Figure 2) use too small text font, that makes almost impossible reading the content.

4.    An important detail for automated image processing is that of standarized illumination, in this way to eliminate an unnecessary variability in the images acquired. Did the authors take illumination as an important factor in their setup?

5.    Instruments and devices, e.g. the computer, the digital camera. Add details of manufacturer and its country.

6.    Python is described as the environment for programming and executing the machine learning algorithm, but where are these algorithms executed?

7.    Line 354. SVM 520?

8.    Supplementary info. I cannot understand Table S4. Each sample 1, 2, 3 and 4 has the 8 rice varieties?

Author Response

Response to Reviewer 2:

Thank you for your time and valuable suggestion. Your suggestions have been incorporated in manuscript and highlighted with green color.

Comment 1: Revised manuscript describes an artificial vision system to assess quality of rice grain. The system is formed by a digital camera controlled from a Raspberry Pi miniature computer and a machine learning identification algorithm, capable of identifying eight rice varieties, in a clear use of intelligent system against food fraud. The final algorithm for identification was Random forests, chosen by comparison from a set of seven different algorithms. Final performance achieved is an identification accuracy of 77.0%; a similar algorithm can be used for determining the value price of adulterated rice samples. Application described is interesting.

Comment 1: Abbreviations: a good idea may be to list the abbreviations in alphabetical order

Response: List of abbreviations is added before references section in alphabetical order.

Comment 2: Introduction needs to comment on Near-infrared methods, which are essentially equivalent to a camera based technique, and can work by direct observation of the solid sample.

Response: Line 45-49, suggestion has been incorporated.

Comment 3: Figures incorporating text (Figure 1, Figure 2) use too small text font, that makes almost impossible reading the content.

Response: The quality of Figure 1 and Figure 2 is enhanced.

Comment 4: An important detail for automated image processing is that of standarized illumination, in this way to eliminate an unnecessary variability in the images acquired. Did the authors take illumination as an important factor in their setup?

Response: Dear reviewer, we have considered illumination as an important factor in our setup. Details regarding illumination has been mentioned in revised manuscript.

“The complete experimental setup is placed in a closed chamber and multiple LED lights placed at uniform distance to reduce illumination.”

Comment 5: Instruments and devices, e.g. the computer, the digital camera. Add details of manufacturer and its country.

Response: The details of the instruments has been added in the revised manuscript.

Comment 6: Python is described as the environment for programming and executing the machine learning algorithm, but where are these algorithms executed?

Response: The capturing of images is done in the camera module, however the post-processing of the images is done in the Desktop (Dell.Inc, i7 processor, 64 GB Ram and Nvidia GPU).

Comment 7: Line 3­54. SVM 520?

Response: Correction has been done.

Comment 8: Supplementary info. I cannot understand Table S4. Each sample 1, 2, 3 and 4 has the 8 rice varieties?

Response: Dear Reviewer, the Supplementary Table-4 shows the Four samples of eight different rice grains predicted using ML algorithm.

For the validation, 50 grains of high quality ’Basmati’ rice variety is mixed with 30 grains of low quality Tibar Basmati (TB) variety. Subsequently, image is captured, and results are obtained. Each of the four samples consist of higher priced and lower priced rice grains in a fixed ratio of 5:3. This is carried to ensure fair comparison of model performance on each of the four samples. The validation results of the 4 rice samples are detailed in Supplementary Table S4. Also, the average price factor per grain is calculated based on the price of each predicted rice grain. Since we don’t know the weight of each grain, we will use the average length of each type to obtain the price factors. This is compared with the actual price factor of the mixture, and the price factor without adulteration for each sample.

Reviewer 3 Report

Background information should include first level of seasonal variation of rice grains characteristic according to weather and technology. This has to be outlined, because it is very important in relation to accuracy of the model and the scope of data needed to train machine. Second, a point of big data management is not presented. So, in the background some examples of big data portal (to have enough data to increase accuracy) and its common use should be presented.

The limitation of research method and design has to be explained in more details.

The future research direction is missing.

I suggest to compare ML and fuzzy system – to see precision and costs and applicability. I also suggest to give some explanation how this ML model will be used in industry context.

The following sentence I suggest to change:

(42-43)The chemical constituents are determined using various chemical procedures which are labo- rious, expensive, and destructive. – I suggest change: The chemical constituents are determined using various lab procedures that take time, raise costs, have certain ecological foot and need specific knowledge.

Author Response

Response to Reviewer 3:

Dear reviewer, thank you for your valuable response. Manuscript has been revised as per your suggestions and changes are highlighted with sky blue colour.

Comment 1: Background information should include first level of seasonal variation of rice grains characteristic according to weather and technology. This has to be outlined, because it is very important in relation to accuracy of the model and the scope of data needed to train machine. Second, a point of big data management is not presented. So, in the background some examples of big data portal (to have enough data to increase accuracy) and its common use should be presented.

Response: Dear reviewer, thank you for valuable suggestion. The background information regarding seasonal variation of rice grain is added in introduction section. Dear reviewer, we have not used the Bigdata algorithms for machine learning model, however, this research can be implemented in future scope, hence we have added the following lines in the future scope.

“Moreover, Big data can be used here to train the Machine learning models, image enhancement, image classification and segmentation, leading to the increase in model efficiency”.

Comment 2: The limitation of research method and design has to be explained in more details.

Response: We are comparing different Machine Learning models for training and identification purposes and the model shows the maximum efficiency of 77%. With training using Bigdata the efficiency of the model may further be improved. This is the research limitation and may be implemented in futuristic research.

Comment 3: The future research direction is missing.

Response: Suggestion incorporated in conclusion section.

Comment 4: I suggest to compare ML and fuzzy system – to see precision and costs and applicability. I also suggest to give some explanation how this ML model will be used in industry context.

Response: Dear reviewer, we initially started solving this problem using set of rules (such as fuzzy logic). For the rice grain it becomes difficult to define manual rules for different varieties of rice grain. Hence, we have implemented the ML algorithms to solve this problem.

Fuzzy systems we struggled to model complex relationships between variables in an image processing problem. This is because the rules and membership functions used in fuzzy systems are often based on simple for if-else statements. Fuzzy systems are difficult to interpret, especially as they become more complex.

Comment 5: The following sentence I suggest to change:

(42-43) The chemical constituents are determined using various chemical procedures which are laborious, expensive, and destructive. – I suggest change: The chemical constituents are determined using various lab procedures that take time, raise costs, have certain ecological foot and need specific knowledge.

Response: Suggestion incorporated.

Reviewer 4 Report

The authors attempted to develop an approch based on ML to predict rice quality using geometric and color features. it is an interesting topic.

However, I have some issues that should be addressed before publication. the literature review is weak. the authors should improve the literature review to clarify the originality of the presented manuscript.

some equations were presented in the results and discussion section. these equations can be in the M and M section.

the results are well-written and presented. However, the discussion is not presented. please improve the discussion of the results with regard the practical implication of the proposed approach at industrial scale. What are the limitations of the proposed method? please highlight the future works to improve the proposed methodology.

Author Response

Response to Reviewer 4:

Dear reviewer, thank you for your time and valuable suggestion. We have revised manuscript as per your suggestion and changes are highlighted with pink colour.

Comment 1: The authors attempted to develop an approch based on ML to predict rice quality using geometric and color features. it is an interesting topic.

However, I have some issues that should be addressed before publication. the literature review is weak. the authors should improve the literature review to clarify the originality of the presented manuscript.

Response: Suggestion incorporated

Comment 2: some equations were presented in the results and discussion section. these equations can be in the M and M section.

Response: Dear reviewer, this is in-fact a very good suggestion to not present any mathematical relation in the results. Thank you very much.

We have changed the section name as model comparison and Results, conclusion and future scope are discussed in the end.

Comment 3: the results are well-written and presented. However, the discussion is not presented. please improve the discussion of the results with regard the practical implication of the proposed approach at industrial scale. What are the limitations of the proposed method? please highlight the future works to improve the proposed methodology.

Response: Dear reviewer, we have revised the conclusion section and added discussion into it. The model may further be launched with a mobile application to evaluate the actual price of the adulterated rice available in local market.

Round 2

Reviewer 2 Report

Comment 5 - I only detected a minor mislead. Authors provide a LED iluminated chamber, I suppose to increase, improve and standarize illumination, not to reduce it. Please correct.

Author Response

Dear Reviewer

Thank you for your observation. Correction has been done as per your suggestion.

Reviewer 4 Report

the paper was improved. 

Author Response

Dear Reviewer,

Thank your for your Kind response for revised manuscript.